# A Deep Learning-Based Electromagnetic Signal for Earthquake Magnitude Prediction

**DOI:** 10.3390/s21134434

**Published:** 2021-06-28

**Authors:** Zhenyu Bao, Jingyu Zhao, Pu Huang, Shanshan Yong, Xin’an Wang

**Affiliations:** 1The Key Laboratory of Integrated Microsystems, Peking University Shenzhen Graduate School, Shenzhen 518055, China; 1901213009@pku.edu.cn (Z.B.); yongshanshan@pku.edu.cn (S.Y.); anxinwang@pku.edu.cn (X.W.); 2School of Electrical Automation and Information Engineering, Tianjin University, Tianjin 300072, China; jingyu_zhao210@tju.edu.cn; 3School of Information Science and Engineering, Northeastern University, Shenyang 110819, China; 4Engineering Department, Shenzhen MSU-BIT University, Shenzhen 518172, China

**Keywords:** earthquake magnitude prediction, electromagnetic sensor, deep learning, data augmentation

## Abstract

The influence of earthquake disasters on human social life is positively related to the magnitude and intensity of the earthquake, and effectively avoiding casualties and property losses can be attributed to the accurate prediction of earthquakes. In this study, an electromagnetic sensor is investigated to assess earthquakes in advance by collecting earthquake signals. At present, the mainstream earthquake magnitude prediction comprises two methods. On the one hand, most geophysicists or data analysis experts extract a series of basic features from earthquake precursor signals for seismic classification. On the other hand, the obtained data related to earth activities by seismograph or space satellite are directly used in classification networks. This article proposes a CNN and designs a 3D feature-map which can be used to solve the problem of earthquake magnitude classification by combining the advantages of shallow features and high-dimensional information. In addition, noise simulation technology and SMOTE oversampling technology are applied to overcome the problem of seismic data imbalance. The signals collected by electromagnetic sensors are used to evaluate the method proposed in this article. The results show that the method proposed in this paper can classify earthquake magnitudes well.

## 1. Introduction

Earthquakes are one of the significant natural disasters facing human society which not only cause heavy casualties but also serious economic losses. In fact, monitoring earthquake plays a very important role for the early detection and warning of earthquake events. It is beneficial to provide crucial information for earthquake response in advance through classifying earthquake disaster levels.

Simple and traditional methods are based on the study of precursory phenomena before earthquakes or historic earthquake data analysis [1]. The precursor phenomena includes groundwater level change, TEC change, seismic quiescence, anomalous electromagnetic field changes, and abnormal animal behavior [2]. Lakkos et al. proposed a back-propagation (BP) neural network to predict earthquake magnitude [3]. Zhang et al. predicted a large earthquake and a number of aftershocks with seismic precursors including seismic quiescence and change in the vertical component of geomagnetism [4]. However, this method uses the earthquakes which have already occurred to train and verify their network, which are taken from a relatively small area (20.0° E–21.5° E, 37.5° N–40.0° N), and the accuracy of the test set does not exceed 80% [3,4]. Studies based on historical data analysis date as far back as 1939 and continue to be pursued today [5,6,7,8,9]. The experimental results from these researches have demonstrated the effectiveness of their method with precursory phenomena and historical earthquake records. In summary, traditional methods are simple and highly interpretable, but these simple features may fail to discover and fully utilize some hidden information contained in seismic data.

Compared with the above methods, deep learning (DL) could predict earthquakes without explicitly modeling certain features. This has led a growing number of scholars to research DL. Jae-Mo Kang proposed a novel ground vibration monitoring scheme for MEMS sensed data by exploiting the DL technique. [10] Mousavi designed a network consisting of convolutional and recurrent layers for magnitude estimation [11]. Dario Jozinovic applied a CNN model to predict the magnitude of ground motions [12]. Perol et al. introduced ConvNetQuake to detect local micro-seismic earthquakes according to signal waveforms. They also show that ConvNetQuake performs well in other earthquake data set [13]. Lomax et al. used CNN to quickly characterize the location, magnitude and other parameters of the earthquake [14]. S. Mostafa Mousavi investigated CNN–RNN to quickly detect weak signals in order to identify earthquakes [15]. Ratiranjan Jena studied a CNN network to assess the probability of earthquakes in the Indian subcontinent [16]. In the same year, Ratiranjan researched another CNN model to assess the magnitude and damage of earthquakes in Indonesia [17]. Subsequently, J. A. Bayona presented two world-famous seismic models in order to assess seismic hazards [18]. The experimental results have demonstrated that certain implicit features might solve the earthquake prediction problem from another perspective. Although DL algorithms can make full use of the hidden information contain in earthquakes, they lack interpretability in the theoretical system.

In short, most of the existing works only use either the explicit features of earthquakes defined or extracted by geologists or experts, or the implicit features (e.g., features vector) extracted by DL methods. The two methods may have a problem of information loss [19]. In order to achieve more accurate earthquake prediction, an effective and potential method combining the advantages of both explicit and implicit features is investigated in this paper.

According to fault theory, the occurrence of an earthquake will cause the plates to squeeze each other, and further cause changes in the electromagnetic field near the epicenter [20]. In this paper, an inductive electromagnetic sensor is firstly introduced to acquire ground vibration information. The sensor adopts a laminated magnetic core to increase its effective area and reduce eddy current loss. At the same time, the effective magnetic permeability is improved by establishing magnetic flux collectors at both ends of the magnetic core. In this way, the sensitivity of the sensor can be greatly improved. In addition, a magnetic negative feedback technology is proposed, which breaks through the limitation of the resonance frequency point, broadens the bandwidth of the sensor monitoring signal, and solves the phenomenon of phase mutation in the frequency band [21]. Then, a CNN model is proposed to classify the earthquake magnitude according to the data of the inductive electromagnetic sensor. In order to obtain the deep connection between sensed data and seismicity, a High-Dimensional-Feature-Extraction block and Temporal-Correlation block were designed. In addition, considering the imbalanced samples, noise simulation technology and SMOTE over-sampling technology were used to augment samples. Finally, extensive data captured from the proposed inductive electromagnetic sensor are used to evaluate the model. The experiment results show that the CNN model demonstrates good performance in earthquake magnitude prediction and has an accuracy which can reach 97.88%.

## 2. Electromagnetic Sensor

Electromagnetic technology represents an important branch in the study of earthquake precursors. Compared with other research methods such as geology, geophysics, and geochemistry, there is a fast response in electromagnetic detection technology which can respond to crustal movement several days, or even hours, before [22,23,24]. In fact, the measurement depth of the electromagnetic sensor is directly related to its frequency band range, and the sensitivity of the sensor is also inseparable from the measurable range of seismic magnitude [25,26,27,28]. Therefore, it is important to design an electromagnetic sensor with broadband, high sensitivity, and low noise for earthquake monitoring.

According to Faraday’s law of electromagnetic induction, the induced voltage value of the sensor is determined by the signal frequency, the effective permeability of the magnetic core, the effective cross-sectional area, and the number of coils:(1)$$e=2\pi fNS_{0}\mu_{a}B$$
where e is the induced voltage output by the sensor, B is the magnetic flux density, and f, N, $S_{0}$ and $\mu_{a}$ are the signal frequency, the number of turns of the coil, the effective cross-sectional area of the core, and the effective permeability of the core, respectively. The sensitivity of the sensor is defined as the ratio of the induced voltage to the magnetic induction intensity:(2)$$\frac{e}{B}=2\pi fNS_{0}\mu_{\alpha }$$

It can be seen from the definition of sensitivity that sensor sensitivity is affected by the cross-sectional area, magnetic core permeability, and frequency. Therefore, the sensitivity of the electromagnetic sensor is directly affected by the properties of the induction coil and the magnetic core.

### 2.1. Effective Permeability Analysis with Magnetic Flux Collector

According to [20], electromagnetic waves are generated when earthquakes occur. A new type of electromagnetic sensor structure was proposed by Acoustic and Electromagnetics to Artificial Intelligence (AETA) which is used to detect changes in electromagnetic waves in underground space. Thus, it can be utilized for seismic signal detection [29,30]. The sensor adds two magnetic flux collectors on both sides of the magnetic core to increase the $\mu_{a}$ value of the magnetic core. The structure is shown in Figure 1. The magnetic flux collector is made by high-permeability materials, and the distribution of the magnetic field is also concentrated in the core, which are beneficial to reduce reluctance and improve the effective permeability. The finite element simulation software Comsol is used to simulate the proposed inductive electromagnetic sensor with a magnetic core size of 0.03 m × 0.020 m × 0.550 m, and the sensitivity is shown in Figure 2. The results show that the new sensor structure with a higher effective magnetic permeability can improve the sensitivity of the sensor.

### 2.2. Research on the Effective Area of Laminated Cores

The initial sensitivity of the sensor is proportional its area. In fact, the magnetic flux of the coil near the magnetic core is larger than the magnetic flux of the coil far from the magnetic core, which means the latter is negligible. In addition, the magnetic flux is much denser on the surface due to the skin effect, which further reduces the effective area of the core.

In order to solve the problem, the magnetic core is made of materials with low conductivity and high permeability. On the one hand, the high permeability magnetic core can make the magnetic field signal more concentrated and further improve the sensitivity of the sensor. One the other hand, the magnetic core is insulated to reduce conductivity, which can, in turn, reduce the eddy current loss and improve the signal-to-noise ratio of the signal. The structure of magnetic core is shown in Figure 3.

### 2.3. Research on Magnetic Negative Feedback Technology

There is a resonance point with the phase mutation phenomenon in the coil which can be known by analyzing its equivalent circuit. In this paper, magnetic negative feedback technology was proposed to solve this problem. As shown in Figure 4, the induction coil will generate an induced electromotive force and output a voltage signal due to the alternating magnetic field. A magnetic field opposite to the measured magnetic field will be generated on the feedback coil after the output voltage is amplified by the amplifier, thereby forming a negative feedback of magnetic flux to the measured magnetic field. The output signals of the induction coil, whose frequency is near the resonance frequency, will be compensated by negative feedback. The negative feedback makes the amplitude-frequency characteristic curve of the sensor in related frequency band flat.

After introducing the magnetic negative feedback technology, the simulation curve of the amplitude–frequency characteristic is shown in Figure 5. It can be seen that the above technology solves the problem of sudden changes in frequency and phase at the resonance point. The method not only greatly improves the bandwidth, but also has flat amplitude–frequency characteristics in the frequency band.

## 3. CNN Networks

Conventional methods are based on studying precursory phenomena before earthquakes and taking advantage of previous experience in seismological research to recognize any anomalous behavior. However, these methods have a higher FPR (False Positive Rate) [31]. Benefitting from the enhancement of processing speed and computing power of computer, machine learning (ML) is used for earthquake predicting due to its advantages in classification and prediction fields. Support Vector Machine (SVM) is a typical ML method for classification, which maximizes the interval between positive and negative samples on the training set by finding the best separation hyper-plane in the feature space [28]. In terms of magnitude prediction, some features extracted from the p-wave and s-wave, such as slope (M), independent term (B), and correlation coefficient (R), are recorded by the seismograph. These features are regarded as the input of SVM to classify the magnitude, which can achieve a delightful result [32]. Rouet-Leduc et al. proposed that the ML method could identify some unknown information or signals from seismic wave signals, and can help to predict earthquakes better than conventional methods [33]. However, it has been found that the detection and extraction of effective features in traditional ML methods is not only time-consuming, but also requires extensive professional knowledge. In fact, it is always affected by the increasing complexity of real-world problems. In contrast, DL can assemble simple features through a variety of nonlinear transformations containing billions of weight parameters and further abstract more complex features. The DL net gradually optimizes these parameters during the training process so as to establish an incomprehensible mapping from input to output [34]. Therefore, it has been widely used in earthquake monitoring with a great performance in recent years [35,36]. In fact, DL had been used to extract deep information, which may cause some information loss [19]. Thus, we propose a method for extracting both shallow features and deep features. On the one hand, it extracts shallow features from raw data to construct 3D feature-map as shown in Figure 6. On the other hand, the 3-D map is used as the input of the DL model to further extract deep features for classification.

### 3.1. Shallow Features Extract

Here, raw data are typical time series data. Considering the following characteristics, namely large data volume, low single-point information density, and obvious periodicity, it is significant to extract useful features from the raw data as the input of CNN. It can not only improve the training speed, but also get data mapping in different dimensions. The acquired shallow features mainly include time domain features, frequency domain features, and wavelet transform features.

The time-domain features are first extracted. Specifically, the time-domain features include mean $\mu_{s}$, variance $\sigma_{s}$, Absolute maximum $Max_{s}$, Power $P_{s}$, Skewness $S_{s}$, Kurtosis $K_{s}$ and short-time energy $E_{s}$, as shown in (3)–(8), respectively.
(3)$$\mu_{s}=\frac{1}{T}\sum_{t=1}^{T}s\left(t\right)$$
(4)$$\sigma_{s}=\frac{1}{T}\sum_{t=1}^{T}{\left(s\left(t\right)-\mu_{s}\right)}^{2}$$
(5)$$P_{s}=\frac{1}{T}\sum_{t=1}^{T}{\left(s\left(t\right)\right)}^{2}$$
(6)$$K_{s}=\frac{\frac{1}{T}\sum_{t=1}^{T}{\left(s\left(t\right)-\left(\mu_{s}\right)\right)}^{4}}{{\left(\frac{1}{T}\sum_{t=1}^{T}{\left(s\left(t\right)-\left(\mu_{s}\right)\right)}^{2}\right)}^{2}}$$
(7)$$S_{s}=\frac{\frac{1}{T}\sum_{t=1}^{T}{\left(s\left(t\right)-\left(\mu_{s}\right)\right)}^{3}}{{\left(\frac{1}{T}\sum_{t=1}^{T}{\left(s\left(t\right)-\left(\mu_{s}\right)\right)}^{2}\right)}^{\frac{3}{2}}}$$
(8)$$E_{s}=\sum_{m=n-\left(N-1\right)}^{n}{\left(s\left(m\right)\right)}^{2}$$
where s(t), t=1,2,3,…, represents raw electromagnetic data, and T is the signal length, generally equal to 30,000. The n refers to time n, and N is the window length in (8).

In the frequency domain, the frequency spectrum of the signal is mainly obtained through a Fourier transform, and the power values of different frequency bands are recorded. Considering that the electromagnetic radiation of seismic activity will be lost during the process of underground propagation, the higher the frequency the greater the corresponding loss. At the same time, the antenna radiation power increases with the quadratic power of the electromagnetic wave frequency, and much of the high-frequency signal is lot. As a result, the monitoring frequency bands for seismic radiation anomalies are generally ultra-low frequency (ULF) monitoring, which also covers extremely low frequency (ELF) and very low frequency (VLF) monitoring. When analyzing data in this article, we mainly use low-frequency and ULF data collected by AETA equipment. In fact, the specific frequency bands are 0~5 Hz, 5~10 Hz, 10~15 Hz, 15~20 Hz, 20~25 Hz, 25~30 Hz, 30~35 Hz, 35~40 Hz, 40~60 Hz, 140~160 Hz and other frequency bands. Moreover, each frequency spectrum is described from the center of gravity frequency, the mean square frequency, the frequency variance, and the spectrum entropy.

In terms of wavelet transform, db4 is used as the wavelet base to perform 6-layer wavelet decomposition due to its better effect in processing rock sound [37]. According to characteristics of electromagnetic disturbance, it is essential to pay attention to the ultra-low frequency range. The ultra-low frequency (below 30 Hz) band information in the detection signal will be more noticed for electromagnetic disturbance data. The sampling rate of the electromagnetic probe is 500 Hz. Thus, we used the reconstructions details 4~6 and approximation 6. The frequency ranges of the details are 15.63~31.25 Hz, 7.81~15.63 Hz and 3.91~7.81 Hz, respectively, and the range of approximation is 0~3.91 Hz. After that, we extracted four statistical features (mean, variance, maximum, and power) from each new wavelet using the same calculation process as the time domain features. The frequency band of details and approximations are expressed by the following two equations:(9)$$\frac{f_{s}}{2^{j+1}}\sim \frac{f_{s}}{2^{j}}$$
(10)$$0\sim \frac{f_{s}}{2^{j+1}}$$
where j is the number of decomposition of vibration signals, and $f_{s}$ is the sampling frequency. In summary, a total of 51 features of the original electromagnetic disturbance signal are extracted, as shown in the Table 1.

### 3.2. The Model Structure

DL has been widely used in various fields recently. In the method of classification, it has been demonstrated that CNN is more efficient than other approaches, especially training samples. This motivated us to apply it to seismic magnitude classification.

Lin et al. proposed a 1 × 1 convolution layer in their work first [38]. Szegedy et al. used parallel multi-scale-convolution filters to obtain different information of the input image in their GoogleNet structure, which could obtain a better deep information of the image [39]. K. He et al. proposed a residual block, which significantly increased the depth and width of the network while keeping the computation workload at a reasonable level [40]. Inspired by their researches, we used the above-mentioned structures of CNN to find the temporal information of each shallow feature and correction-information between different shallow features.

Figure 7 describes the model structure proposed in this paper. The first part is High-Dimensional-Feature-Extraction (HDFE) block, and its input feature is a 3D feature-matrix. A Temporal-Correlation block is the second part, which consists of four convolution units. A convolution unit is a convolution layer followed by a BN layer and a Max-Pooling layer. The last part is Classification block, which includes of a ‘bottleneck’ unit and a classification unit with a 6-way soft-max layer. A ‘bottleneck’ unit comprises three convolution units.

In the HDFE block, inspired by the inception network structure [39], a four-way convolution is used to find the correlation information between shallow features. Filters of different sizes can extract higher-level features in electromagnetic disturbance signal. The structure of these convolution units is shown in Figure 7a. The 1 × 1 convolutions are added before 3 × 3 and 5 × 5 convolution in order to achieve the dimension reduction modules. This setting can approximate a local sparse structure of CNN by using readily available dense matrices [41]. After the above four-way convolution process is completed, all output from different convolution units is parallelly merged, which retained the unique characteristics of each group of filters. The architecture of HDFE is employed and is written as:(11)$$B_{1}=H\left(\left[Con\left(B_{0}\right),Con_{1}\left(Con\left(B_{0}\right)\right),Con_{2}\left(Con\left(B_{0}\right)\right),Con\left(Con\left(B_{0}\right)\right)\right]\right)$$
where $B_{1}$ is the new feature map after concatenating, $B_{0}$ is the input map and Con represents the convolution function. Moreover, the length of the output $B_{1}$ should keep constant with $B_{0}$. Thus, convolution in 3×3 and 5×5 requires 1×1 and 2×2 padding with the size (r−1)/2, respectively. The application of dense connection provides a short-cut path for the gradients during backpropagation and alleviates the vanishing-gradient problem for the training of the neural network [39].

Next, a convolution operation is performed at the output of concatenation of the inception units in the Temporal-Correlation block. This block contains 4 convolutional units. Each convolutional unit is constituted of a convention layer, a BN layer, and a Max-Pooling layer. The input of this block is a 3D feature-map with a plane size of 27 × 24 on a single channel, which refers to 27 days and 24 h, respectively. The model could find time-related information at the same time on different days or at different times on the same day through convolution and pooling operations on the two dimensions of the feature map. In short, the investigated net-work can extract high-dimensional time-related features and robust features from the fused features through the HDFE block [42].

After that, a simplified deep residual networks (DRN) was added with a residual block because the DRN has excellent performance in classification for image recognition [40]. The architecture of DRN was employed and can be described as:(12)H(x)=f(x)+x
where x and H(x) are the input and output of the DRN, respectively. The left part of Figure 7c presented the architecture of the DRN, and the residual block is made up by the convolution layer, BN layer, Max-Pooling layer, and ReLU layer. There are two dense layers converting the output of the DRN into classification results.

Compared to single CNN, the DRN has a higher training speed and easier gradient transmission [40]. Thus, we transposed the data matric into the DRN layers, which can both improve the final accuracy and make full use of features of the current layer and the previous layer. Finally, the typical soft-max function was used for multi-classification by transforming the output of the last layer to conditional probabilities of a label. The label with the highest probability was decided as the final classification result of the Modified CNN net. The cross-entropy function characterized in Equation (13) was adopted as the loss function of this model.
(13)$$H\left(p,q\right)=-\sum_{i}p\left(i\right)\log\left(q\left(i\right)\right)$$
where q(i) is the estimated distribution and p(i) is the true distribution. In addition, the adaptive moment estimation optimization with initial learning rate 0.0001 was used to minimize the loss function. Particularly, dropout technology was applied to avoid over fitting problem [43].

### 3.3. Data Set

Since the magnitude classification task involves complex parameters such as time and space, it is difficult to set a unified label manually or by modeling. Considering the rationality of the label setting, this article uses the data collected every 27 days to judge whether there will be an earthquake on the 28th day. In this way, the complicated magnitude classification task is simplified, and the earthquake information of the next day can be known in advance. The definition is described in Figure 8. The sample was collected from 1 January 2017 to 1 January 2021. A total of 6936 samples were obtained through data set balancing process, and the data set distribution ratio was balanced. Among them, there are 1040 training samples and 5896 verification samples. The method of data set balance processing will be introduced in Section 3.4.

At the same time, earthquake magnitude prediction can be regard as a regression task because the degree of damage varies with the magnitude of the earthquake. This article is divided into six levels according to the magnitude of the earthquake, as shown in Table 2.

### 3.4. Over-Sampling Data

In fact, the amount of seismic data is much smaller than the amount of non-seismic data, so there is a problem with respect to unbalanced data samples. The most straightforward and common approach in dealing with imbalance of seismic data is sampling balance, which consists of both over-sampling and under-sampling. The former enhances the minority class by accurately copying the examples of the minority class, while the latter randomly takes away some examples of the majority class [44]. For example, Min Ji et al. adopted a random over-sampling method to overcome the imbalance problem [45]. The current paper divides the labels into six categories, and there is also an imbalance problem. Under-sampling techniques results in a huge loss of valid information, which is not appropriate in constructing a robust model of magnitude prediction. Thus, an over-sampling method was applied for the minority samples. The currently popular over-sampling techniques are random re-sampling [46], SMOTE [47], adding-white-noise and data generated by GAN. Considering that SMOTE and adding-white-noise technology can generate simulation data quickly and efficiently, these methods were applied. SMOTE is an over-sampling method, which created new synthetic samples along the line between the minority samples and their selected nearest neighbors, as shown in (14):(14)$$x_{n}=x_{i}+rand\left(0,1\right)\times \left(x_{k}-x_{i}\right),$$
where $x_{n}$ is new synthetic minority samples, $x_{i}$ is the randomly selected minority examples, $x_{k}$ are their selected nearest neighbors, and rand(0,1) is a random number greater than zero and less than one. In fact, SMOTE treats all minority samples equally and does not take account of the boundary regions, which could lead to poor performance [48]. To overcome this problem, the Borderline–SMOTE was applied to achieve better predictions. Borderline–SMOTE is a method based on SMOTE. It only over-samples or strengthens the borderline minority examples. For one thing, it finds out the borderline minority samples. For another, generating synthetic data from it and added the new data to the original data set. We define the whole data set as T, the minority class as P and the majority class as N:(15)$$P=\left\{p_{1},p_{1},\ldots ,p_{pnum}\right\},N=\left\{n_{1},n_{1},\ldots ,n_{nnum}\right\}$$
where pnum and nnum are the number of minority and majority samples. We calculate its nearest neighbors k from the whole data set T for each $p_{i}\left(i=1,2,\ldots ,pnum\right)$ in the class P. If these nearest neighbors k all belong to class N, the $p_{i}$ is considered invalid and ignored. Nevertheless, if the number of these k neighbors belonging to class N is more than the number belonging to class P, the $p_{i}$ is put into a set DANGER. In contrast, the $p_{i}$ will be put into another set SAFE while the number of these k neighbors belonging to class N is less than the number belonging to class P. When a new synthetic sample is generated later, only the data in the set DANGER is used and the set SAFE is ignored. Based on the above analysis, the Borderline–SMOTE algorithm can be classified into Borderline–SMOTE-1 and Borderline–SMOTE-2. For each data $d_{i}\left(i=1,2,\ldots ,dnum\right)$ in the set DANGER, Borderline–SMOTE1 selects its m(0≤m≤k) nearest neighbors from set P, and attains m×dnum new synthetic examples by formula (12). Borderline-SMOTE2 selects its m nearest neighbors from the whole set T, regardless of whether this neighbor belongs to class P or belongs to class N. Another different point is that rand(0,1) in formula (12) is changed to rand(0,0.5) in the calculation of Borderline–SMOTE2, so the generated synthetic sample is closer to the minority class in DANGER. According to [48], the former could better improve the performance of the classifier, so we decided to use the Borderline–SMOTE1 algorithm to expand the data set. Moreover, the earthquakes with a relatively high magnitude usually occupy a small part of the vast majority of seismic data, while most of the earthquake events in the dataset are small-scale. Thus, we further use the method of adding Gaussian noise to simulate the measurement noise of the sensor and expand the data set to overcome the problem of data imbalance. Due to the special distribution of real data, adding white noise according to the value range of the real data can make the increased data distribution closer to the true distribution of the original data. For the selection of Gaussian noise parameters, the current work tried a total of nine combinations of different mean values (0, 0.5, 1) and different variances (0.1, 0.5, 1). It was found that when the mean value is 0 and the variance equal to 0.5, the model obtains the best performance.

## 4. Experiments

### 4.1. Model Setup and Hyper-Parameters

The inputs of the current model are 6936 feature-maps with size of 51 × 27 × 24 introduced in Chapter 3. We divided 6936 samples into training set and test set after shuffling, the proportions are 85% and 15%, respectively.

The networks in this article all use the same training configuration and hyper-parameters. The model is implemented using Pytorch 1.6.0 and deployed on GeForce RTX 2080 Ti GPU. And before data input into each layer, we normalized the data of each channel after flattening, as follows:(16)$$\frac{x_{i}-\mu_{i}}{\sigma_{i}}\left(i=1,2,\ldots ,N\right)$$
where $x_{i}$ is the flattened tensor when the channel dimension equals i, and N, $\mu_{i}$ and $\sigma_{i}$ are the length, mean and standard deviation of $x_{i}$, respectively. An Adam optimizer was applied with default values of parameters recorded in [49], and the mini-batch size was set as 128. The training data were shuffled for every epoch, and each network was trained for enough epochs about 200. The learning rate was initialized to 0.01, and we used the learning rate optimization function named Reduce LR On Plateau in the Pytorch library, which would reduce the learning rate when the loss of training set stops falling. The weight decay was set to 0.005, and dropout at 0.5 for all layers except those layers in classification block. The model would stop training data when the loss of validation set does not decrease for consecutive 100 generations.

### 4.2. Loss and Accuracy

We set the maximum number of iterations to 150. It can be seen from the loss curve and accuracy curve of the training set and the validation set that the model achieves the best performance after the 70th iteration in Figure 9. At the beginning of the iteration, compared to the validation set, the loss and accuracy curve of training set have smaller fluctuations. The reason may be that the model is constantly looking for the local optimization during training, and the validation set is equivalent to the generalization process of the training set. Therefore, those small fluctuations may be amplified on the verification set, which appear as a dramatically changes shown in Figure 9. Moreover, it is easy to see that the verification set loss no longer decreases and the model stabilizes after 70 iterations.

### 4.3. Algorithm Comparison and Discussions

Classification accuracy (ACC), F1 score (F1), Precision (Pre), and Recall were used to evaluate the performance of our proposed model. The parameters of F1 score, Precision, and Recall were set to macro-average for multiclass task. The results of the current CNN net using six types of magnitude labeling schemes are listed in Table 3.

In this section, other typical ML and DL methods using the same 3D feature-map for earthquake magnitude classification are also constructed. The results of the algorithms are presented in Table 4. ML methods including SVM with linear kernel, SVM with rbf-kernel, Decision, Random Decision Forest, and KNN. Specially, these methods all use Grid search and cross validation to acquire the most suitable parameters. All compared classifiers are implemented by the Scikit-learn toolkit. In order to obtain the best results of different classifiers, the feature data are normalized into 0 and 1 before they are input to the classifiers. On the other hand, we used some DL algorithms in the computer vision field as a contrast, such as LSTM, CNN + biLSTM, Resnet50, Resnet101, Vgg16, Vgg19 and Nasnet. The results of these algorithms are also shown in Table 4.

It is easy to see that the algorithm proposed in this paper achieves the best results in the test set, while Resnet101 has the shortest time consumption. Compared with DL algorithms, ML methods generally have a slower processing speed, are more time-consuming, and have lower accuracy. Particularly, CNN + LSTM is a net similar to the current method, which adds a unit of Bi-LSTM after the DRN in the third part of the current article proposed. This method not only reduces the training speed, but also demonstrates poor performance. This may be due to the fact that the time extraction module in this paper has processed the time dimension information well and the output of the time module no longer carries the time dimension information.

## 5. Conclusions

The influence of earthquake disasters on human social life is positively related to the magnitude and intensity of the earthquake. Casualties and property losses can be effectively avoided by accurately predicting earthquakes. In this study, an electromagnetic sensor, which was used to collect earthquake signals, was investigated in order to assess earthquakes in advance. The sensor adopted a laminated magnetic core to increase its effective area and reduce eddy current loss. At the same time, the effective magnetic permeability was improved by establishing magnetic flux collectors at both ends of the magnetic core. In this way, the sensitivity of the sensor was greatly improved. In addition, a magnetic negative feedback technology was proposed, which breaks through the limitation of the resonance frequency point, broadens the bandwidth of the sensor monitoring signal, and solves the phenomenon of phase mutation in the frequency band. Then, a CNN model was proposed to classify the earthquake magnitude according to the data of the inductive electromagnetic sensor. In order to obtain the deep connection between sensed data and seismicity, both a High-Dimensional-Feature-Extraction block and Temporal-Correlation block were designed. In addition, considering the imbalanced samples, noise simulation technology and SMOTE over-sampling technology were used to augment samples. Finally, extensive data captured from the proposed inductive electromagnetic sensor were used to evaluate model. The experiment results show that the CNN model has good performance in earthquake magnitude prediction, and that its accuracy can reach 97.88%. The results demonstrated that the proposed method can be regarded as a powerful tool for the above-mentioned sensor to improve the performance of magnitude classification. The method of combining spatial and temporal information in the form of shallow features can also be tried on data collected by other devices.

## Figures and Tables

**Figure 1 sensors-21-04434-f001:**
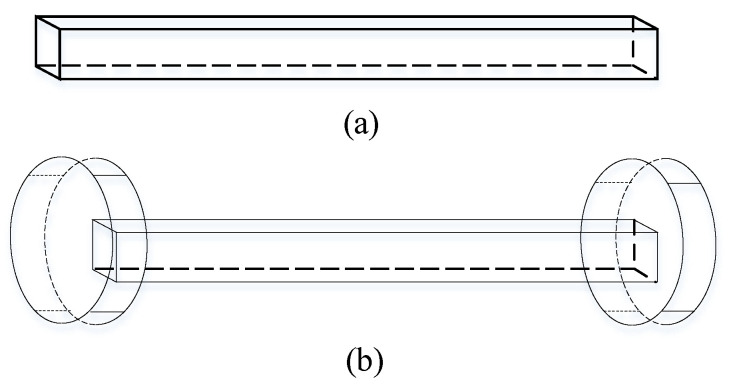
Shape of the core: (**a**) rectangular core and (**b**) rectangular core with flux concentrators.

**Figure 2 sensors-21-04434-f002:**
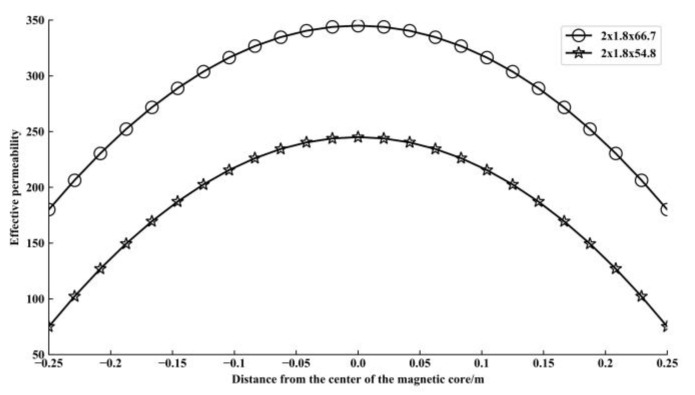
The simulation curves of the apparent permeability distribution for the rectangular cores with different sized flux concentrators.

**Figure 3 sensors-21-04434-f003:**
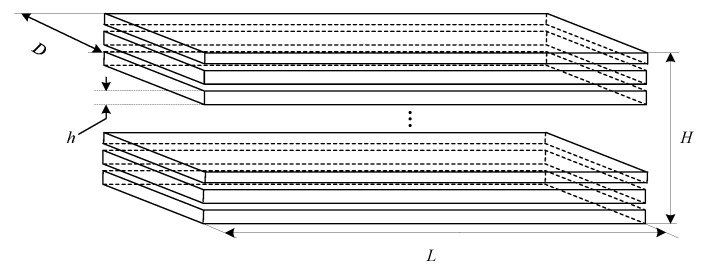
Schematic diagram of laminated magnetic core.

**Figure 4 sensors-21-04434-f004:**
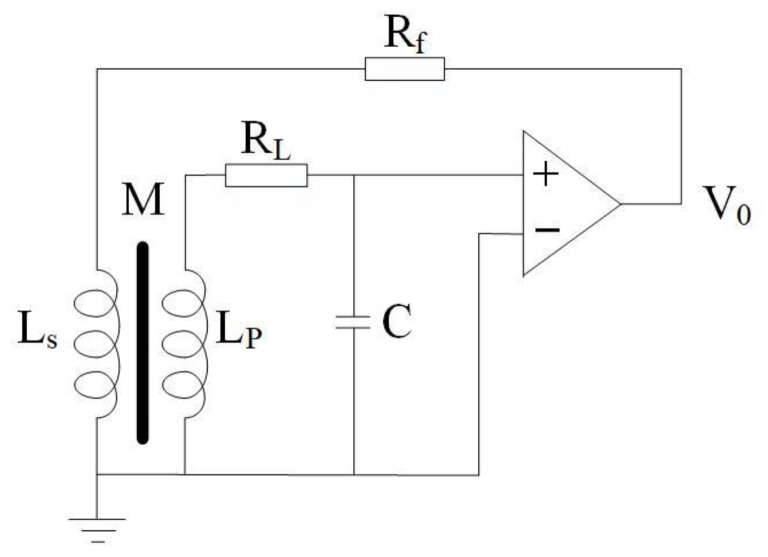
Schematic diagram of negative feedback technology.

**Figure 5 sensors-21-04434-f005:**
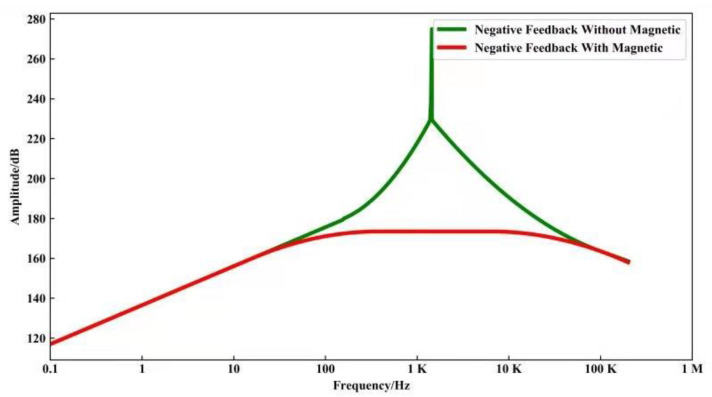
The simulation curve of the amplitude–frequency characteristic.

**Figure 6 sensors-21-04434-f006:**
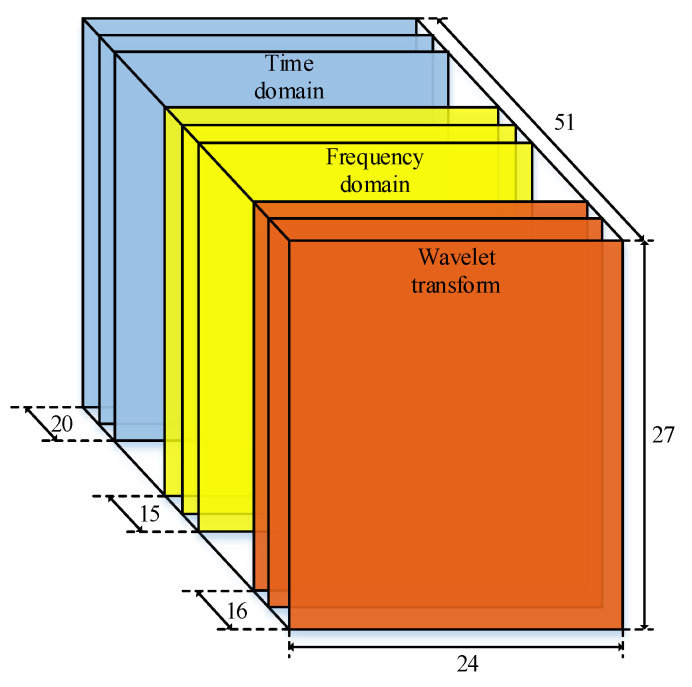
The 3D shallow feature map.

**Figure 7 sensors-21-04434-f007:**
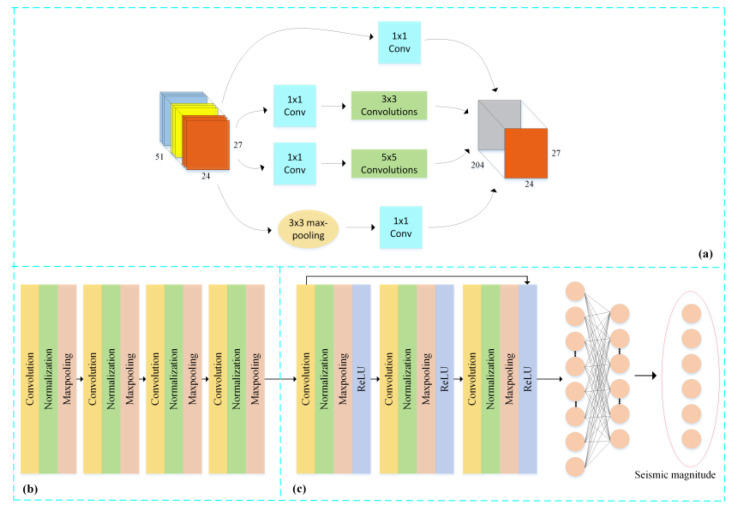
The detailed structure of our model. (**a**) High-Dimensional-Feature-Extraction block, (**b**) Temporal-Correlation block, and (**c**) Classification block.

**Figure 8 sensors-21-04434-f008:**
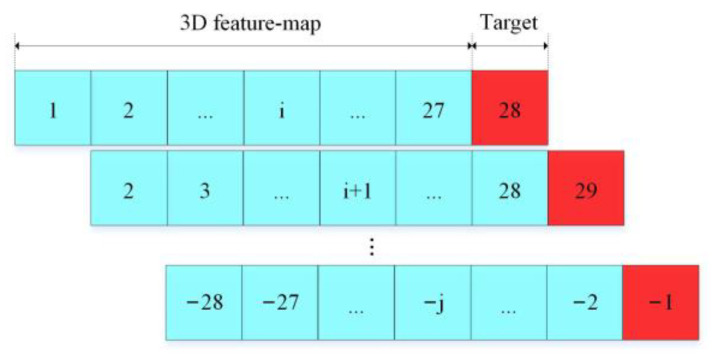
Definition of classification target.

**Figure 9 sensors-21-04434-f009:**
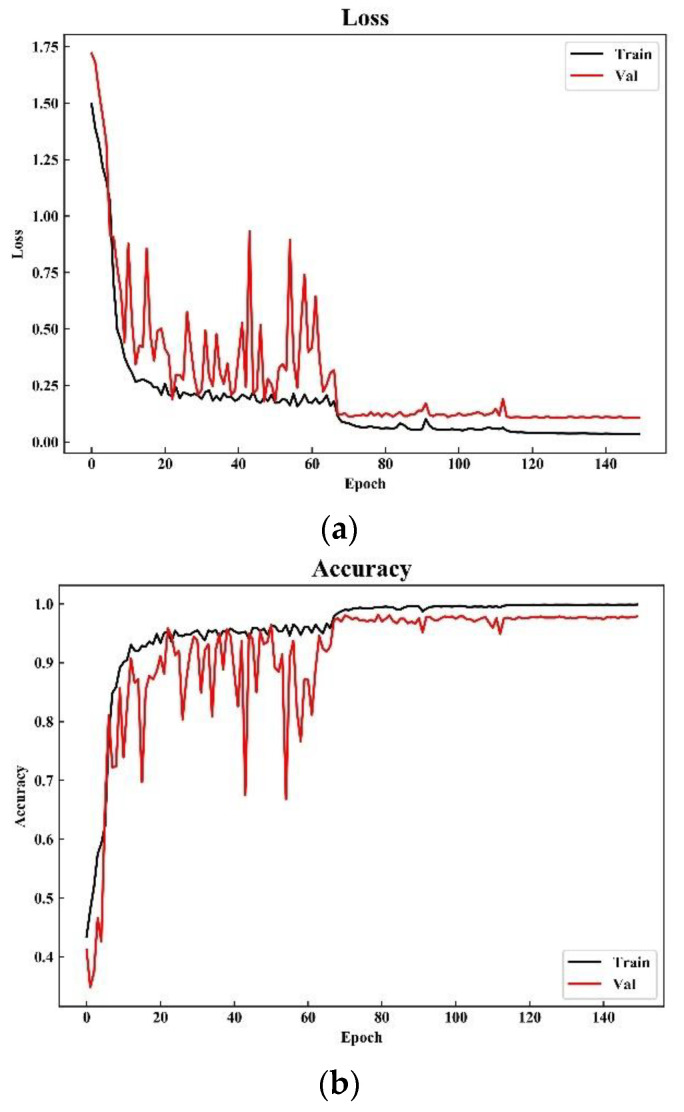
Loss and accuracy of training and validation set: (**a**) loss and (**b**) accuracy.

**Table 1 sensors-21-04434-t001:** The extracted shallow 51 features.

Index	Feature Description
1	Variance
2	Power
3	Skewness
4	Kurtosis
5	Maximum absolute value
6	Mean absolute value
7	Absolute maximum 5% position
8	Absolute maximum 10% position
9	Short-term energy standard deviation
10	Maximum short-term energy
11	0~5 Hz power
12	5~10 Hz power
13	10~15 Hz power
14	15~20 Hz power
15	20~25 Hz power
16	25~30 Hz power
17	30~35 Hz power
18	35~40 Hz power
19	40~60 Hz power
20	140~160 Hz power
21	Power ratio of other frequency bands
22	Center of gravity frequency
23	Mean square frequency
24	Frequency variance
25	Frequency entropy
26	Mean value of absolute value of level 4 detail
27	Level 4 detail energy
28	Maximum energy value of level 4 detail
29	Level 4 detail energy value variance
30	Mean value of absolute value of level 5 detail
31	Level 5 detail energy
32	Maximum energy value of level 5 detail
33	Variance of Level 5 detail energy value
34	Mean value of absolute value of level 6 detail
35	Level 6 detail energy
36	Maximum energy value of level 6 detail
37	Level 6 detail energy value variance
38	Approximate mean value of absolute value at level 6
39	Level 6 approximate energy
40	Maximum approximate energy value of level 6
41	Level 6 approximate energy value variance
42	Mean absolute value of ultra-low frequency
43	Variance of ultra-low Frequency
44	Ultra-low frequency power
45	Ultra-low frequency skewness
46	Ultra-low frequency kurtosis
47	Maximum absolute value of ultra-low frequency
48	Maximum 5% position of absolute value of ultra-low frequency
49	Maximum 10% position of absolute value of ultra-low frequency
50	Ultra-low frequency short-term energy standard deviation
51	Maximum ultra-low frequency short-term energy

**Table 2 sensors-21-04434-t002:** Labels.

Magnitude Range (M.)	Label
0 < M. < 3.5	0
3.5 < M. < 4	1
4 < M. < 4.5	2
4.5 < M. < 5	3
5 < M. < 6	4
M. > 6	5

**Table 3 sensors-21-04434-t003:** Model Evaluation.

M.	Pre	Recall	F1
0 < M. < 3.5	0.948571	0.927374	0.937853
3.5 < M. < 4	0.955056	0.988372	0.971429
4 < M. < 4.5	0.970588	0.988024	0.979228
4.5 < M. < 5	0. 975802	0.983425	0.981643
5 < M. < 6	0. 989385	0.988166	0.984048
M. > 6	0. 993163	0.991362	0.993048
Macro-average	0.979036	0.979227	0.979034

**Table 4 sensors-21-04434-t004:** Model Comparison.

Model	Accuracy	Time Consuming(s)
SVM	0.934	10,457
Decision Tree	0.8687	22,236
KNN	0.8691	23,330
Random Forests	0.7592	9657
LSTM	0.7493	6154
CNN + LSTM	0.8903	6800
Resnet50	0.9324	1386
Resnet101	0.9182	1001
Vgg16	0.9086	2261
Vgg19	0.9162	2464
Nasnet	0.9353	1841
Current Method	0.9788	1736

## Data Availability

The data presented in this study are available from the authors upon reasonable request.

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
