# Peer review of "A Deep Learning-Based Electromagnetic Signal for Earthquake Magnitude Prediction"

_sensors, 2021, doi:10.3390/s21134434_

Round 1

Reviewer 1 Report

In my opinion the article “A Deep Learning-Based Electromagnetic Signal for Earthquake Magnitude Prediction” needs a major revision. It is presented as a work on a method to classify the magnitude of earthquakes by means of a Deep Learning approach but it turns that the proposed method relies on the simulated data of a novel electromagnetic detector proposed by the authors.

The approach is very naive because neither the scientific topics, concerning earthquakes and geophysics, nor the instrumental aspects, concerning detector response and application, are treated with a proper level of deepening. Therefore, the proposed Deep Learning method looks like a sort of ideal exercise completely detached from the critical aspects of a real-world sensor or method devoted to investigate a delicate topic like earthquakes. In this sense, their claim that “the proposed method can be regarded as a powerful tool to improve the performance of magnitude classification” is not acceptable since no real data from electromagnetic sensors are used in this work for test or comparison.

In my opinion, the authors have two choices:

1. They could work both on the scientific and the instrumental aspects concerning this topic with experts of earthquakes, geophysics, magnetosphere physics, plasma physics and developers of electromagnetic instruments to provide their work with a more solid background on these aspects. In this case, the proposed Deep Learning method would be more robust.

2. They could remove the part concerning the design of the electromagnetic sensor and present they Deep Learning method as an approach for classifying the magnitude of earthquakes by means of an ideal electromagnetic sensor. In this case, this work should be submitted to a different journal.

Here are some remarks on the paper which could help the authors in improving their work.

L. 40 “Although they have achieved good accuracy, in actual predictions, some earthquakes are

still not predicted.” - Please specify with respect what parameters the mentioned accuracy is achieved. Currently, there are no scientifically recognized methods or techniques which can be used for earthquake forecast. This statement is misleading and it must be rephrased or removed.

L. 61 “The experimental results demonstrated that implicit features can solve the earthquake prediction problem from another perspective.” - Actually, it’s not demonstrated. Please use “might solve” instead.

L. 71 “According to fault theory, the occurrence of an earthquake will cause the plates to squeeze each other, and further cause changes in the electromagnetic field near the epicenter.” - Please cite one or more references.

L. 73 “An inductive electromagnetic sensor is firstly introduced to acquire ground vibration information in this paper. The sensor adopts laminated magnetic core to increase effective area and reduce eddy current loss.” - Since the deep learning approach presented in this article would rely on the data produced by the proposed Inductive Electromagnetic Sensor, the title should be “A Deep Learning-Based Electromagnetic Sensor for Earthquake Magnitude Prediction”.

L. 90 - “Electromagnetic technology is an important branch in the study of earthquake precursors. Compared with other research methods, such as geology, geophysics and geochemistry, there is a fast response in electromagnetic detection technology, which can respond to the crustal movement several days or even hours before.” - Please cite one or more reference to justify this statement.

L. 113 “A new type of electromagnetic sensor structure is proposed.” - The authors must explain if this novel detector has been designed for ground or space application as well as as the type of environment it is intended to monitor. The authors must keep in mind that the design of this kind of detector is driven on the type of application they are intended for.

L. 135 “Then multiple sheets of surface insulation are stacked to a magnetic core” – Please specify the material identified as insulator.

L. 156 “The method not only greatly improves the bandwidth, but also has flat amplitude-frequency characteristics in the frequency band.” - Typo (mehod instead of method) - At the present stage it’s an ideal detector. Therefore the predicted response is unrealistic because it does not take into account all instrumental effects which can degrade its signal and the overall performance. This point must be underlined in this discussion.

L. 163 “… the abnormal behavior.” - “… the anomalous behavior.” would be better.

L. 163 “However, these methods had a higher FPR (False Positive Rate).” - Please explain the origin of this higher FPR or cite a reference.

L. “T is the signal length, generally equal to 30000.” - Please explain why this value is acceptable.

L. 210 “In fact, the specific frequency bands are 0~5Hz, 5~10Hz, 10~15Hz, 15~20Hz, 20~25Hz, 25~30Hz, 30~35Hz, 35~40Hz, 40~60Hz, 140~160Hz and other frequency bands.” - Please justify from the scientific or technical point of view the choice of this low frequency range since the expected range of the proposed instrument is up 100 kH (Figure 5). In recent years, missions devoted to the study of electromagnetic anomalies and/or earthquake precursors (DEMETER, Swarm, CSES, ...) have been focused on a larger frequency range (up to a few MHz), so the proposed range is very small.

L. 216 “The ultra-low frequency (below 30Hz) band information in the detection signal will be more noticed for electromagnetic disturbance data.” - Please explain why the ULF band is more suitable to detect signal or cite a reference. Moreover, cite that real ULF data are affected by some known disturbances called Schumann resonances at 7.83 Hz (fundamental), 14.3, 20.8, 27.3 and 33.8 Hz to be kept into account.

L. 218 “The sampling rate of the electromagnetic probe is 500Hz, thus we used the reconstructions details 4~6 and approximation 6.” - Please explain the motivations behind the choice of this spcific value of the sampling rate.

L. 354 “Thus, we further use the method of adding Gaussian noise to simulate the measurement noise of the sensor and expand the data set to overcome the problem of data imbalance.” - Please describer the criterion and/or the parameters used to simulate the expected noise of the sensor.

L. 436 “The results demonstrated that the proposed method can be regarded as a powerful tool to improve the performance of magnitude classification.” - The authors should rephrase this statement and underline that this method relies on the proposed sensor design. They cannot infer that the proposed method will work on data produced by other electromagnetic sensors.

Author Response

  1. The approach is very naive because neither the scientific topics, concerning earthquakes and geophysics, nor the instrumental aspects, concerning detector response and application, are treated with a proper level of deepening. Therefore, the proposed Deep Learning method looks like a sort of ideal exercise completely detached from the critical aspects of a real-world sensor or method devoted to investigate a delicate topic like earthquakes. In this sense, their claim that “the proposed method can be regarded as a powerful tool to improve the performance of magnitude classification” is not acceptable since no real data from electromagnetic sensors are used in this work for test or comparison.

Reply 1: First of all, I am glad that you took the time to review my paper. Since electromagnetic anomalies are one of the important anomalous signals for earthquake precursors, there are already many mature electromagnetic seismic sensor devices in the world, such as the German GMS-07. Compared with these mature magnetic sensor devices, it has the advantages of small size, low cost, high precision and intensive deployment. In other words, our equipment is easier to obtain large amounts of data, which is suitable for today's big data era of high-performance computing. In addition, the design of our sensor's electromagnetic probe adopts magnetic flux negative feedback technology, which makes the transmission characteristics of the sensor smooth at the resonance frequency. It not only broadens the bandwidth of the magnetic signal collection, but also improves its robustness. The specific details of the design are described in detail in the second section. Therefore, we are not utilizing a kind of simulation or ideal data, and the original data comes from the electromagnetic sensor.

  1. L. 40 “Although they have achieved good accuracy, in actual predictions, some earthquakes are still not predicted.” - Please specify with respect what parameters the mentioned accuracy is achieved. Currently, there are no scientifically recognized methods or techniques which can be used for earthquake forecast. This statement is misleading and it must be rephrased or removed.

Reply 2: Modify the expression in the original as follows:

However, the above method uses the earthquakes that have occurred to train and verify their network, which are taken from a relatively small area, and the accuracy of the test set does not exceed 80% [3, 4].

[3]. Lakkos, S.; Hadjiprocopis, A.; Comley, R.; Smith, P. A neural network scheme for Earthquake prediction based on the seismic. Proceedings of the IEEE Workshop on Neural Networks for Signal Processing, Ermioni, Greece, 1994; IEEE: New York, USA.

[4]. Yue, L.; Yuan, W.; Yuan, L.; Zhang, B.; Wu, G. Earthquake prediction by RBF neural network ensemble. Proceedings of the International Symposium on Neural Networks, Dalian, China, 2004; Springer, Berlin, Heidelberg.

  1. L. 61 “The experimental results demonstrated that implicit features can solve the earthquake prediction problem from another perspective.” - Actually, it’s not demonstrated. Please use “might solve” instead.

Reply 3: Modify the expression in the original as follows:

“The experimental results demonstrated that implicit features might solve the earthquake prediction problem from another perspective.”

  1. L. 71 “According to fault theory, the occurrence of an earthquake will cause the plates to squeeze each other, and further cause changes in the electromagnetic field near the epicenter.”- Please cite one or more references.

Reply 4: Add the reference as follows:

[Added 1] Reid, H. F., The California earthquake of April 18, 1906, The mechanics of the earthquake, Report of the (California) State Earthquake Investigation Commission, vol. 2, Publ. 87, pp. 1 – 192, Calif. State Earthquake Invest. Comm., Sacramento, 1910.

  1. L. 73 “An inductive electromagnetic sensor is firstly introduced to acquire ground vibration information in this paper. The sensor adopts laminated magnetic core to increase effective area and reduce eddy current loss.” - Since the deep learning approach presented in this article would rely on the data produced by the proposed Inductive Electromagnetic Sensor, the title should be “A Deep Learning-Based Electromagnetic Sensor for Earthquake Magnitude Prediction”.

Reply 5: Thank you for your suggestion. But the deep learning method proposed by this work is more dependent on the electromagnetic signals collected by the equipment, we hope to use the original title.

  1. L. 90 - “Electromagnetic technology is an important branch in the study of earthquake precursors. Compared with other research methods, such as geology, geophysics and geochemistry, there is a fast response in electromagnetic detection technology, which can respond to the crustal movement several days or even hours before.” - Please cite one or more reference to justify this statement.

Reply 6: Add the references as follows:

[Added 2] Fraser-Smith, A. C.; Bernardi, A.; McGill, P. R.; Ladd, M. E.; Helliwell, R. A.; Villard, O. G. Low-frequency magnetic field measurements near the epicenter of the Ms 7.1 Loma Prieta Earthquake. Geophysical Research Letters 1990, 17, 1465-1468.

[Added 3] Hirano, T.; Hattori, K. ULF geomagnetic changes possibly associated with the 2008 Iwate-Miyagi Nairiku earthquake. Journal of Asian Earth Sciences 2011, 41, 442–449.

[Added 4] Gokhberg, M. B.; Morgounov, V. A.; Yoshino, T.; Tomizawa, I. Experimental measurement of electromagnetic emissions possibly related to earthquakes in Japan. Journal of Geophysical Research: Solid Earth 1982, 87, 7824-7828.

  1. L. 113 “A new type of electromagnetic sensor structure is proposed.” - The authors must explain if this novel detector has been designed for ground or space application as well as as the type of environment it is intended to monitor. The authors must keep in mind that the design of this kind of detector is driven on the type of application they are intended for.

Reply 7: It is used for ground detection. According to [Added 1], electromagnetic waves are generated when earthquakes occur. The designed sensor is used to detect changes in electromagnetic waves in underground space, so it can be used for seismic signal detection.

[Added 1] Reid, H. F., The California earthquake of April 18, 1906, The mechanics of the earthquake, Report of the (California) State Earthquake Investigation Commission, vol. 2, Publ. 87, pp. 1 – 192, Calif. State Earthquake Invest. Comm., Sacramento, 1910.

  1. L. 135 “Then multiple sheets of surface insulation are stacked to a magnetic core” – Please specify the material identified as insulator.

Reply 8: The magnetic core is made of materials with low conductivity and high permeability. On the one hand, the high permeability magnetic core can make the magnetic field signal more concentrated, and further improve the sensitivity of the sensor. One the other hand, the magnetic core is insulated to reduce the conductivity, which can reduce the eddy current loss and improve the signal-to-noise ratio of the signal.

  1. L. 156 “The method not only greatly improves the bandwidth, but also has flat amplitude-frequency characteristics in the frequency band.” - Typo (mehod instead of method) - At the present stage it’s an ideal detector. Therefore the predicted response is unrealistic because it does not take into account all instrumental effects which can degrade its signal and the overall performance. This point must be underlined in this discussion.

Reply 9: With such a structure, the signal-to-noise ratio of the sensor can be improved, and the amplitude-frequency characteristic curve can also be optimized. We have verified through theoretical analysis that the proposed sensor is superior to traditional electromagnetic sensors, which has been mentioned in my previous answers. We made corresponding sensors based on this structure to detect earthquakes, and the detected signals are real noisy signals.

  1. L. 163 “… the abnormal behavior.” - “… the anomalous behavior.” would be better.

Reply 10: We agree to this modification, thank you for your suggestion.

  1. L. 163 “However, these methods had a higher FPR (False Positive Rate).” - Please explain the origin of this higher FPR or cite a reference

Reply 11: Add the reference as follows:

[Added 5]. Kamigaichi, O.; Saito, M.; Doi, K.; Matsumori, T.; Tsukada, S.; Takeda, K.; Shimoyama, T.; Nakamura, K.; Kiyomoto, M.; Watanabe, Y. Earthquake Early Warning in Japan: Warning the General Public and Future Prospects. Seismological Research Letters 2009, 80, 717-726.

  1. “T is the signal length, generally equal to 30000.” - Please explain why this value is acceptable.

Reply 12: Since the sampling rate is 500Hz, we choose one minute of continuous raw data to calculate the characteristic value, the signal length is generally equal to 30000.

  1. L. 210 “In fact, the specific frequency bands are 0~5Hz, 5~10Hz, 10~15Hz, 15~20Hz, 20~25Hz, 25~30Hz, 30~35Hz, 35~40Hz, 40~60Hz, 140~160Hz and other frequency bands.” - Please justify from the scientific or technical point of view the choice of this low frequency range since the expected range of the proposed instrument is up 100 kHz (Figure 5). In recent years, missions devoted to the study of electromagnetic anomalies and/or earthquake precursors (DEMETER, Swarm, CSES, ...) have been focused on a larger frequency range (up to a few MHz), so the proposed range is very small.

Reply 13: Considering that the electromagnetic radiation of seismic activity will be lost during the process of underground propagation, the higher the frequency, the greater the corresponding loss. At the same time, the antenna radiation power increases with the quadratic power of the electromagnetic wave frequency, the high-frequency signal lost a lot. As a result, the monitoring frequency bands for seismic radiation anomalies are generally ultra-low frequency (ULF) monitoring, which also covers extremely low frequency (ELF) and very low frequency (VLF) monitoring. When analyzing data in this article, we mainly use low-frequency and ULF data collected by equipment.

  1. L. 216 “The ultra-low frequency (below 30Hz) band information in the detection signal will be more noticed for electromagnetic disturbance data.” - Please explain why the ULF band is more suitable to detect signal or cite a reference. Moreover, cite that real ULF data are affected by some known disturbances called Schumann resonances at 7.83 Hz (fundamental), 14.3, 20.8, 27.3 and 33.8 Hz to be kept into account.

Reply 14: In Reply 13, it has been mentioned why low frequency and ULF are used. As for the Schumann resonance due to global thunderstorms, it generally has stable frequency domain parameters and spectrum structure. In recent years, studies have found that these inherent parameters will be disturbed before an earthquake occurs, and ELF waves are used in the field of earthquake monitoring [added 6].

[added 6]. Gazquez, J. A.; Garcia, R. M.; Castellano, N. N.; Fernandez-Ros, M.; Perea-Moreno, A. J.; Manzano-Agugliaro, F. Applied Engineering Using Schumann Resonance for Earthquakes Monitoring. Applied Sciences 2017, 7.

  1. L. 218 “The sampling rate of the electromagnetic probe is 500Hz, thus we used the reconstructions details 4~6 and approximation 6.” - Please explain the motivations behind the choice of this spcific value of the sampling rate.

Reply 15: We focus on the frequency range of 0~30Hz, but we find that there is still a weak magnetic field in the frequency range of 30Hz~200Hz when we do actual field monitoring. And the magnetic field disturbance in the frequency band above 200Hz almost disappears. Therefore, we set the sampling rate to 500Hz to facilitate the later analysis of the data in the 0~200Hz frequency range.

  1. L. 354 “Thus, we further use the method of adding Gaussian noise to simulate the measurement noise of the sensor and expand the data set to overcome the problem of data imbalance.” - Please describer the criterion and/or the parameters used to simulate the expected noise of the sensor.

Reply 16: We decided to add the following at the end of line 358:

For the selection of Gaussian noise parameters, the current work tried a total of nine combinations of different mean values (0, 0.5, 1) and different variances (0.1, 0.5, 1). It was found that when the mean value is 0 and the variance equal to 0.5, the model obtains the best Excellent performance.

  1. L. 436 “The results demonstrated that the proposed method can be regarded as a powerful tool to improve the performance of magnitude classification.” - The authors should rephrase this statement and underline that this method relies on the proposed sensor design. They cannot infer that the proposed method will work on data produced by other electromagnetic sensors.

Reply 17: Modify the expression in the original as follows:

The results demonstrated that the proposed method can be regarded as a powerful tool for the above-mentioned sensor to improve the performance of magnitude classification. The method of combining spatial and temporal information in the form of shallow features can also be tried on data collected by other devices.

Reviewer 2 Report

This paper proposed a new type of electromagnetic sensor structure, which is used to collect earthquake signals. The authors proposed a CNN and designed a 3D feature-map, which can be used to solve the problem of earthquake magnitude classification by combining the advantages of shallow features and high-dimensional information. However, there are a number of critical issues in the paper, which should be clearly addressed before publication, as follows.

  1. The method to design a low noise electromagnetic sensor is not explained in the sentence “Therefore, it is important to design an electromagnetic sensor with broadband, high sensitivity and low noise for earthquake monitoring.”

  1. Some hyper-parameters related to CNN network are not described in this paper. The paper only mentioned how to choose the learning rate and batch size, but other hyper-parameters such as decay rate, drop-out rate are also important. More detailed information should be provided here.

  1. The number of samples in Chapter 3 is 6936, but the number of samples in Chapter 4 is 7009. What’s more, the proportion of the training set and test set are also different. The reason should be added in the context.

  1. There are many methods about over-sampling as mentioned in this paper, but the methods adopted by the authors in the paper are not clear. For example, the Borderline- SMOTE algorithm can be classified into BordelineSMOTE-1 and BorderlineSMOTE-2. However, it is unclear which one is chosen and why. Moreover, it will be better to describe the initial amount of data, the amount of data each method raises individually, and also details are needed.

  1. Why the curve in Figure 9 changes dramatically at the beginning of the iteration? Please explain it.

  1. The performance of the six magnitude labels are showed in Table 3. It's clear that the larger magnitude of the label, the better the performance it is. However, this phenomenon is not explained in the article.

Author Response

  1. The method to design a low noise electromagnetic sensor is not explained in the sentence “Therefore, it is important to design an electromagnetic sensor with broadband, high sensitivity and low noise for earthquake monitoring.”

  1. Some hyper-parameters related to CNN network are not described in this paper. The paper only mentioned how to choose the learning rate and batch size, but other hyper-parameters such as decay rate, drop-out rate are also important. More detailed information should be provided here.

 L. 375 The weight decay set to 0.005, and dropout at 0.5 for all layers except those layers in classification block.

  1. The number of samples in Chapter 3 is 6936, but the number of samples in Chapter 4 is 7009. What’s more, the proportion of the training set and test set are also different. The reason should be added in the context.

change 7009 in chapter4 to 6936

  1. There are many methods about over-sampling as mentioned in this paper, but the methods adopted by the authors in the paper are not clear. For example, the Borderline- SMOTE algorithm can be classified into BordelineSMOTE-1 and BorderlineSMOTE-2. However, it is unclear which one is chosen and why. Moreover, it will be better to describe the initial amount of data, the amount of data each method raises individually, and also details are needed.

  1. 320 robust model.
  2. 352 According to [42], the former could better improve the performance of the classifier, so we decided to use the Borderline-SMOTE1 algorithm to expand the data set.

  1. Why the curve in Figure 9 changes dramatically at the beginning of the iteration? Please explain it.

  1. 381: Because the learning rate optimization function are used to find the suitable learning rate during the model fitting process, the verification set curve fluctuates a little. 

At the beginning of the iteration, compared to the validation set, the loss and accuracy curve of training set have smaller fluctuations. The reason may be that the model is constantly looking for the local optimization during training, and the validation set is equivalent to the generalization process of the training set. Therefore, those small fluctuations may be amplified on the verification set, which appear as a dramatically changes shown in Fig. 9.

  1. The performance of the six magnitude labels are showed in Table 3. It's clear that the larger magnitude of the label, the better the performance it is. However, this phenomenon is not explained in the article.

  1. 394: It is easy to find that the categories corresponding to higher magnitudes have better performance. According to our analysis, this may be because higher-level earthquakes generally have more abnormal data, which is higher or lower characteristic value when mapped to the feature-map matrix or there are some more special rules to be discovered by the network. On the contrary, for lower-level earthquakes, especially the earthquakes corresponding to label 0, which are generally stable or general in most cases. However, there will be some slight disturbance interference, such as seismic magnitude not exceeding 3.5 or influence exceeding the range of 27 days, which affects the performance of lower-level labels.

Round 2

Reviewer 1 Report

Remarks and proposed corrections/updates in the authors' responses are acceptable but they are not included in the latest version of the article (file sensors-1196041-peer-review-v2.pdf).

For instance, in the authors' response it's stated:

"Reply 2: Modify the expression in the original as follows:

However, the above method uses the earthquakes that have occurred to train and verify their network, which are taken from a relatively small area, and the accuracy of the test set does not exceed 80% [3, 4]."

This modified expression cannot be found in the article.

Similarly, it's stated:

"Reply 3: Modify the expression in the original as follows:

“The experimental results demonstrated that implicit features might solve the earthquake prediction problem from another perspective.”"

The expression in the latest version of the article is still:

"The experimental results demonstrated that implicit features can solve the earthquake prediction problem from another perspective."

An so on for the following points.

Therefore, I recommend the authors to apply all the correction listed in their response as soon as possible in order to make the article acceptable.

Author Response

Dear editors and reviewers:

I am very grateful to your comments for the manuscript which not only enhance our paper but also helpful for our future research. Based on your comments and requests, we have made modification on the original manuscript. 

Thank you very much. We look forward to your positive response.

Sincerely,
Pu Huang

Reviewer 2 Report

The revised manuscript can be accepted.

Author Response

(The authors gave the same response as above.)

Round 3

Reviewer 1 Report

The authors have adopted the proposed requests and recommendations during the review process by improving the overall quality of the article. Therefore, the publication in the present form is recommended.
